# Understanding coverage of antenatal care in Palestine: Cross-sectional analysis of Palestinian Multiple Indicator Cluster Survey, 2019–2020

**Masako Horino** [1,2]*, **Salwa Massad**[3], **Saifuddin Ahmed**[4], **Khalid Abu Khalid**[5], **Yehia Abed**[6]

1 United Nations Relief and Works Agency for the Palestine Refugees in the Near East Department of Health, Headquarters, Amman, Jordan, 2 Center for Human Nutrition, Department of International Health and Sight and Life Global Research Institute, Johns Hopkins Bloomberg School of Public Health, Baltimore, MD, United States of America, 3 Palestinian National Institute of Public Health, Ramallah, Palestine, 4 Department of Population, Family and Reproductive Health, Bloomberg School of Public Health, Johns Hopkins University, Baltimore, MD, United States of America, 5 Palestinian Central Bureau of Statistics, Ramallah, Palestine, 6 School of Public Health, Al-Quds University, Jerusalem, Palestine

* mhorino1@jhu.edu

**Data Availability Statement:** All Palestinian MICS 2019-2020 files are publicly available from the UNICEF database (https://mics.unicef.org/surveys).

## Abstract

### Introduction

Antenatal care is an essential component of primary healthcare, providing opportunities to screen, prevent, and treat morbidity to preserve the health of mothers and offspring. The World Health Organization now recommends a minimum of eight antenatal care contacts, instead of four, which is challenging in countries exposed to political violence and structural disparities in access to social, economic and healthcare resources as exist in Palestine. This study examines the compliance of the recommend standard of antenatal care in Palestine.

### Methods

We analyzed data from the UNICEF's Palestinian Multiple Indicator Cluster Survey (MICS) 2019–2020. The eligible sample consisted of 2,028 women, 15–49 years of age, living in Palestine, on whom data were available on reported antenatal care services received during the most recent pregnancy within the last two years. Outcome variables of interest were the reported frequencies of antenatal care visits, gestational timing of 1st visit, and services received. Potential risk factors were assessed in women attending less than eight versus eight or more antenatal contacts, as recommended by WHO, by estimating prevalence ratios with 95% Confidence Intervals.

### Results

Overall, 28% of women did not meet the WHO's recommendation of eight or more antenatal contacts, varying from 18% in Central West Bank to 33% in South West Bank across the four areas of Palestine (North, Central, and South West Bank and Gaza Strip). Twelve per-cent of women reported having had no antenatal contacts in the 1st trimester, and these

Access to the datasets require registration as a
MICS Data User from UNICEF.

**Funding:** We would like to disclose that the George
G. Graham Professorship Endowment grant (grant
number 1605030010) from the Johns Hopkins
University Fund Center will provide support for the
publication fee.

**Competing interests:** The authors have declared
that no competing interests exist.

women were two- to three-folds more unlikely to meet WHO recommendation of antenatal contacts than mothers who initiated the antenatal contact in the 1ˢᵗ trimester. Women who had less than eight antenatal contacts were generally poorer, higher in parity, lived in North and South West Bank, sought ANC from either doctor or nurse/midwife only, and initiated antenatal contact in 2ⁿᵈ-to-3ʳᵈ trimesters.

## Conclusion

There were considerable socioeconomic and geographic inequalities in the prevalence of not meeting WHO recommended number of antenatal contacts in Palestine, offering the opportunity to inform, improve and continuously reassess coverage of antenatal care.

## Introduction

Adequacy of antenatal care (ANC) is an important indicator for the Reproductive and Maternal Health Dimension of Sustainable Development Goals (SDGs), and specifically to reach the target 3.8, to achieve universal health coverage by 2030 [1]. ANC is defined as routine care provided to a pregnant woman between conception and the onset of labor. ANC is an essential component of basic primary healthcare and offers a platform for delivery of services that are vital to prevent, detect and treat materno-fetal morbidity, and preserve health throughout pregnancy [2–4]. An adequate number of ANC contacts with a skilled provider, initiated early in pregnancy, is crucial for screening modifiable risk-factors, preventing complications and managing pre-existing conditions [5]. Tests that can be performed during ANC contacts, especially an early contact, include screening for genetic and congenital disorders, periconceptional folic acid supplementation to reduce the risk of neural tube defects, preventing or treating iron deficiency anemia and multiple micronutrient deficiencies through multiple micronutrient supplementation, measuring blood pressure for chronic hypertension, and screening for sexually transmitted infections [5]. An adequate number of ANC contacts throughout pregnancy may also provide an opportunity to detect non-communicable diseases such as diabetes or gestational hypertension and to educate mothers-to-be on reducing modifiable lifestyle risk factors such as smoking, alcohol consumption, drug abuse, healthy eating, and occupational and environmental exposures.

In 2016, WHO released a revised ANC guideline, recommending eight ANC contacts throughout pregnancy, with the first contact occurring in the first trimester (up to 12 weeks of gestation) followed by two and five contacts in the second and third trimesters, respectively [6]. A timely first ANC contact has been shown to be associated with an increased total number of ANC contacts and the content of care received [7]. The recommended shift from four to eight ANC contacts highlights the importance of continuous care, starting early in pregnancy, to ensure early screening. During the years preceding its issuance, ANC coverage, defined as the percentage of women aged 15–49 years provided *any* antenatal care by a skilled provider, was estimated to be 85% globally and 77% in the low-income countries [4]. Previous studies have reported several factors associated with adequate antenatal care: initiation of care in the first trimester of pregnancy, residence in an urban area, especially close to an ANC facility, secondary or higher education, small household size (fewer than five members), higher socioeconomic status, lower parity (fewer than four live births), having health insurance, and being married vs single [3, 7–10]. Furthermore, early pregnancy registration, clinical competence of the healthcare provider, and attentive interactions are also thought to be significant predictors

of achieving the WHO recommended number of eight ANC contacts in pregnancy [7, 10]. Globally, the percentage of women who initiate ANC in the first trimester has increased from 41% in 1999 to 59% in 2013. However, there is still a significant disparity in the receipt of any ANC coverage between low-income countries (24%) and high-income countries (82%) [5].

Palestine includes Northern, Central and South areas of the West Bank and the Gaza Strip. In the Gaza Strip, approximately 2.1 million people are housed in a 365 km$^2$ area of land, with an average population density of 5479 persons per square kilometer, making the area one of the most densely populated urban areas in the world [11]. In the West Bank, 2.9 million Palestinian population resides in approximately 5,655 km$^2$ including East Jerusalem [12]. The two million Palestine refugees comprise 44% of the total population in Palestine (~5 million); 63% of these refugees are in the Gaza Strip and 37% are in the West Bank and East Jerusalem [13].

Deteriorating socioeconomic condition in the Gaza Strip reflects the challenges imposed by the blockade and repeated conflicts. For example, since the 7th of October 2023, humanitarian catastrophe has been unfolding in Palestine, with mass destruction of civil infrastructure and health systems, displacement, blockade of food, water and fuel in the Gaza Strip, while in the West Bank, a complete curfew, an increase in violence, and demolition of Palestinian structures have been implemented. Approximately 1.2 million, which is 24.7% of the total population in Palestine, are estimated to be women of reproductive age (15–49 years) with the fertility rate of 3.8 in Palestine; 3.9 in the Gaza Strip and 3.8 in the West Bank [14]. In 2020, ANC services were provided by 475 Ministry of Health clinics, 65 clinics operated by the United Nations Relief and Works Agency for Palestine Refugees in the Near East (UNRWA), 17 private clinics and 192 clinics run by non-governmental organizations (NGOs) [15–17]. Health services were financed through a mixture of taxes, health insurance premiums and co-payments, out-of-pocket payments, local community financial and in-kind donations, and loans and grants from the international community (e.g. UNRWA). Based on the monetary poverty concept, 29.3% of Palestinian households lived below the poverty level in 2017 with the poverty rates of 13.9% in the West Bank and 53% in the Gaza Strip [18]. Palestine continues to experience military occupation, restrictions of the movement imposed by multiple checkpoints, blockade and siege on the Gaza Strip, geographic divisions, and isolation by construction of the Separation Wall, which limits access to healthcare services [19–21].

It is widely reported that prenatal stress due to conflict, malnutrition and poverty has an impact on fetal development, pregnancy complications and pregnancy outcomes [22]. In addition, deteriorated living conditions in Palestine due to repeated exposures to political violence, ongoing siege on the Gaza Strip since 2017, coupled with the high poverty and unemployment rates affect access to ANC [21]. Given the vulnerability of the population in Palestine, ANC faces a major challenge to provide timely, adequate and appropriate care to pregnant women. However, there is little reported national data on the frequency, timing and contents of services received by the women attending ANC, which are provided by multiple agencies, in Palestine. Some studies of health care access conducted in Palestine rely on sub-national data which are not representative of the overall population, and frequently exclude the Gaza Strip, likely reflecting the challenges of collecting data for the entire Palestine due to the geographical separation between the West Bank and the Gaza Strip [21]. The lack of information needed for management of ANC is one of the major difficulties, restricting the capacity to plan and assess performance [20]. Thus, the objectives of this study are, for the four areas of Palestine (Gaza Strip and Northern, Central and South areas of the West Bank), to: 1) summarize sociodemographic characteristics of pregnant women; 2) estimate the proportion of pregnant women, based on data, whose number of ANC contacts were below WHO recommendation (0–7 contacts; ANC0-7) versus those who were compliant with WHO recommendation (8 or more contacts; ANC8+); 3) identify maternal characteristics associated with ANC0-7, which could

further inform targeting antenatal services, and completing WHO recommended number of ANC contacts among women in Palestine.

## Methods

### Research design

We performed a cross-sectional analysis of secondary data, utilizing publicly available data on women of reproductive age (age 15 to 49) who participated in the Palestinian Multiple Indicator Cluster Survey (MICS) 2019–2020 [14]. MICS is an international household survey initiative by UNICEF which collects national level data from households based on multi-stage, stratified cluster sampling design. Within each region of Palestine (Gaza Strip, Northern West Bank, Central West Bank, South West Bank), urban, rural and camp areas sampling frames were defined. For the first stage, primary sampling units (PSUs) were selected from the census enumeration areas (EA) from the 2017 Palestine Census of Population and Housing, which was partially updated in 2019. The second stage involved listing households in each sampled EA, from which samples of households were selected. The data collection started in December 2019 and was concluded in January 2020 [14]. The survey included 2441 women of reproductive age, 15 to 49 years, with available data on reported frequency, content of services, and timing of ANC contacts during a most recent pregnancy within the last two years. After conducting descriptive analysis, 250 women were missing on the number of antenatal contacts during the most recent pregnancy and 197 women who reported >16 antenatal contacts were excluded due to its implausibility. Therefore, our analysis included 2028 women of reproductive age in Palestine.

### Ethical approval

Ethical approval was not needed for this secondary data analysis as MICS data is publicly available and anonymized. However, the survey protocol was originally approved by the Health Media Lab Institutional Review Board Committee in September 2019 [14]. Verbal consent was obtained from each participant by the trained data collector at the time of interview and documented on a paper questionnaire [14]. All respondents were informed that participation is voluntary, and the information is collected anonymously and kept confidential. Additionally, respondents were informed of their right to refuse to answer any questions and to withdraw from the interview at any time without any consequences.

### Main outcome

Our main outcome of interest was the prevalence of mothers who self-reported receiving ANC contacts 0–7 times (ANC0-7), below the WHO recommendation eight ANC contacts, during the last pregnancy within the two years from the time of data collection. The variable was recoded as a binary outcome (ANC0-7 vs. ANC8+). Binary variable was then created for the women who did not have any contacts for ANC in the 1st trimester versus those who had at least one contact for ANC in the 1st trimester. Receipt of basic ANC components during ANC visit was assessed by three services: blood pressure measurement, urine test, and a blood test. Additional variables included the gestational age (in months and weeks) at which the respondent had her first antenatal contact, recoded into months from one to nine and defined initiation of ANC as the first contact occurring in the first, second or third trimesters. The survey also inquired about the type of health professional who provided ANC services, recorded as a doctor, or nurse/midwife.

## Other variables

Demographic information included women's age (<20, 20–29, 30–39, 40–49), education level (none or basic, secondary, higher than secondary), marital status (currently married or not currently married), number of parities (1, 2, 3, 4, >5), refugee status, area of residence (Gaza Strip, Northern West Bank, Central West Bank, and South West Bank), and type of residency (urban, rural or camp). Wealth index was constructed based on the ownership of consumer goods, dwelling characteristics, water and sanitation, and other characteristics related to wealth of households, and was categorized into 5 categories (richest, richer, middle, poorer, poorest) as a composite indicator of wealth [14]. Types of health insurance was recoded into one multinomial categorical variable using dummy variables, and included governmental, UNRWA, private, Israel, and no insurance.

## Statistical analysis

All analyses were completed using a statistical software STATA version 16 (College Station, TX, USA). The design-based modeling with sample weight, clusters and stratum was used to account for multistage, stratified cluster sampling methodology of Palestinian MICS 2019–2020 [14, 23] All estimates were weighted for disproportionate sampling and non-response rates and the variances were adjusted for higher design-effects due to clustering. Descriptive statistics were used to assess the sociodemographic characteristics of women, ANC coverage in terms of frequency, timing, and service components, and a type of ANC providers. To assess the relationships between ANC0-7 and other covariates, the weighted chi-square tests and adjusted Wald tests were used for categorical variables and continuous variables, respectively. Since the prevalence of our outcome, ANC0-7 was 28%, we decided to use a log-binomial model to estimate prevalence ratios, instead of estimating odds-ratios (ORs) with a logistic regression, for simplicity in expressing the excess risks of a high prevalence outcome as risk ratios [24]. Two different types of design-based log-binomial regression were fitted to examine the association between ANC0-7 and selected characteristics of women. First, univariable log-binomial regression adjusting for sample weight, clusters and stratum was performed. Then, a design-based multivariable log-binomial regression model was fitted to adjust for selected covariates that were identified to be the determinants of ANC coverage, based on the previous literature, including women's age, education level, number of parities, wealth index, area of residence, type of ANC provider, and timing of first antenatal contact [3, 7, 9, 10, 25]. However, this log-binomial regression model failed to converge [26]. Therefore, design-based robust multivariable Poisson regression model was fitted to approximate the log-binomial estimates of prevalence ratios [27]. Nonlinearity of age and parity was assessed by including the quadratic terms to the design-based robust multivariable Poisson regression model. The final model including sample weight, clusters, stratum, women's age, education level, number of parities as a continuous variable and its quadratic term, wealth index, region of residence, type of ANC provider and timing of first antenatal contact. The estimates were presented with 95% Confidence Intervals (CIs).

## Results

### Sample characteristics

Table 1 summarizes the characteristics of 2,028 women aged between 15 to 49 whom data were available on reported ANC services received during the most recent pregnancy within the last two years in the Palestinian MICS 2019–2020. The mean age of the participants was 28 years, with the majority between 20 to 29 years (59%). Approximately half of women obtained

**Table 1. Characteristics of women who participated in the 2019–2020 Palestinian Multiple Indicator Cluster Survey (n = 2,028).**

| Variable | | n (%)[a] |
|---|---|---|
| Total | | 2028 (100) |
| Age (years) | | |
| | Mean (Standard Deviation, SD) | 28.0 (5.8) |
| | Median (Interquartile Range, IQR) | 27 (23, 32) |
| | 15–19 | 81 (4.0) |
| | 20–29 | 1196 (59.0) |
| | 30–39 | 678 (33.4) |
| | 40–49 | 73 (3.6) |
| Education | | |
| | Basic or less | 350 (17.3) |
| | Secondary | 719 (35.4) |
| | Higher than secondary | 959 (47.3) |
| Currently married | | 2014 (99.3) |
| Number of parity | | |
| Mean (SD) | | 3 (2) |
| Median (IQR) | | 3 (2, 4) |
| 1 | | 435 (21.5) |
| 2 | | 484 (23.9) |
| 3 | | 396 (19.5) |
| 4 | | 281 (13.9) |
| >5 | | 432 (21.3) |
| Wealth index | | |
| | Richest | 318 (17.0) |
| | Richer | 395 (21.1) |
| | Middle | 369 (19.7) |
| | Poorer | 351 (18.7) |
| | Poorest | 441 (23.6) |
| Refugee status | | 756 (37.3) |
| Region | | |
| | Northern West Bank | 431 (21.3) |
| | Central West Bank | 264 (13.0) |
| | South West Bank | 481 (23.7) |
| | Gaza Strip | 851 (42.0) |
| Location of residence | | |
| | Urban | 1552 (76.6) |
| | Rural | 326 (16.1) |
| | Camp | 149 (7.4) |
| Health insurance | | |
| | Governmental | 804 (39.7) |
| | UNRWA | 737 (36.4) |
| | Private | 42 (2.1) |
| | Israel | 63 (3.1) |
| | Other | 1 (0.0) |
| | No insurance | 380 (18.8) |
| Type of antenatal care provider | | |
| | Doctor only | 1217 (60.1) |

*(Continued)*

**Table 1.**  (Continued)

| Variable | | n (%)[a] |
|---|---|---|
| | Nurse/Midwife only | 123 (6.1) |
| | Both doctor and nurse/midwife | 687 (33.9) |
| Types of antenatal assessments | | |
| | Blood pressure | 1993 (98.3) |
| | Urine test | 1954 (96.3) |
| | Blood test | 1972 (97.2) |
| | All services | 1907 (94.9) |
| Timing of 1st antenatal contact | | |
| | Mean, gestational age, weeks (SD) | 7.3 (5.8) |
| | Median, gestational age, weeks (IQR) | 4 (4, 8) |
| | 1st trimester | 1784 (88.0) |
| | 2nd trimester | 209 (10.3) |
| | 3rd trimester | 35 (1.7) |
| Number of antenatal contacts | | |
| Mean (SD) | | 9 (3.0) |
| Median (IQR) | | 9 (7, 11) |
| 0–7 antenatal contacts (ANC 0–7) | | 573 (28.2) |
| 8 or more antenatal contacts (ANC 8+) | | 1456 (71.8) |

[a] Design-based modeling with sample weight, clusters and stratum was used to estimate weighted numbers and proportions of women.

secondary education or higher (47.3%). Most of the women were married (99.3%), and average number of parities was three. Thirty-seven percent of respondents self-reported to be a refugee, which could include both the registered refugees who hold a refugee registration card issued by UNRWA and non-registered refugees who do not hold such a card. Majority of women lived in urban areas (76.6%), while 16.1% lived in rural areas and 7.4% in the camps. About 40% of women held governmental health insurance, while 36.4% received free health services from UNRWA as a registered Palestine refugee.

## Coverage of antenatal care

Roughly more than 95% of women reported receiving appropriate ANC contents including blood pressure measurement, urine sample and blood sample tests. Twenty-eight percent of women had less than eight antenatal contacts during their last pregnancy, not meeting the WHO recommended number of contacts, and ~13% of women completed their first antenatal contact after the 1st trimester.

## Factors associated with number of antenatal contacts

Table 2 presents the percentage of women with ANC0-7 vs ANC8+ by their characteristics. Women with ANC0-7 were slightly older (28.8 vs 27.7), less likely to be educated, have greater number of parity (3.7 vs 2.9), and lower distribution of wealth index. Among the four areas of Palestine, prevalence of ANC0-7 was the highest in South West Bank (33.3%), followed by Gaza Strip (29.1%), North West Bank (27%) and Central West Bank (18.1%). In addition, women living in rural areas had the highest proportion of ANC0-7 (29.2%), followed by those living in urban areas (28.3%) and in camps (25.6%). Compared to women with ANC8+, those with ANC0-7 were less likely to receive ANC from doctors only (30.2% vs 69.8%), nurse/

**Table 2. Characteristics of women compared by completion of recommended antenatal care contacts, reported in the 2019–2020 Palestinian MICS (n = 2,028)[a].**

| Women's characteristics | | ANC0-7 | ANC8 | p-value[b] |
|---|---|---|---|---|
| | | N (%) unless otherwise indicated | | |
| Age, Mean (SD) | | 28.8 (5.9) | 27.7 (5.7) | 0.004 |
| <20 | | 17 (20.6) | 64 (79.4) | 0.03 |
| 20–29 | | 306 (25.8) | 880 (74.2) | |
| 30–39 | | 225 (33.5) | 447 (66.5) | |
| 40–49 | | 20 (27.8) | 53 (71.8) | |
| Education | | | | 0.10 |
| | None or basic | 111 (31.8) | 237 (68.2) | |
| | Secondary | 212 (29.7) | 501 (70.3) | |
| | Higher than secondary | 246 (25.8) | 705 (74.2) | |
| Number of parities, Mean (SD) | | 3.7 (2.1) | 2.9 (1.9) | <0.001 |
| 1 | | 66 (15.4) | 365 (84.6) | <0.001 |
| 2 | | 117 (24.3) | 363 (75.7) | |
| 3 | | 129 (32.9) | 263 (67.1) | |
| 4 | | 95 (34.0) | 184 (66.0) | |
| >5 | | 160 (37.5) | 268 (62.5) | |
| Wealth index | | | | 0.01 |
| | Richest | 60 (19.0) | 256 (81.1) | |
| | Richer | 111 (28.5) | 280 (71.6) | |
| | Middle | 109 (29.7) | 257 (70.3) | |
| | Poorer | 116 (33.4) | 232 (66.6) | |
| | Poorest | 132 (30.1) | 306 (69.9) | |
| Refugee status | | | | 0.99 |
| Refugees | | 211 (28.2) | 538 (71.8) | |
| Non-refugees | | 356 (28.2) | 905 (71.8) | |
| Area of residence | | | | 0.002 |
| | Northern West Bank | 116 (27.0) | 312 (73.0) | |
| | Central West Bank | 47 (18.1) | 215 (81.9) | |
| | South West Bank | 159 (33.3) | 318 (66.7) | |
| | Gaza Strip | 246 (29.1) | 598 (70.9) | |
| Location of residence | | | | 0.65 |
| | Urban | 435 (28.3) | 1104 (71.7) | |
| | Rural | 95 (29.2) | 229 (70.8) | |
| | Camp | 38 (25.6) | 110 (74.4) | |
| Type of health insurance | | | | 0.44 |
| | Governmental | 225 (28.2) | 573 (71.9) | |
| | UNRWA | 211 (28.9) | 520 (71.2) | |
| | Private | 8 (19.0) | 33 (81.0) | |
| | Israel | 11 (17.7) | 52 (382.3) | |
| | No insurance | 113 (30.1) | 264 (69.9) | |
| Type of antenatal care provider | | | | <0.001 |
| Doctor only | | 365 (30.2) | 842 (69.8) | |
| Nurse/Midwife only | | 52 (42.3) | 71 (57.7) | |
| Both Doctor and Nurse/Midwife | | 150 (22.1) | 530 (77.9) | |
| Timing of 1st antenatal contact | | | | |
| | 1st trimester | 405 (22.9) | 1363 (77.1) | <0.001 |

(*Continued*)

**Table 2.** (Continued)

| Women's characteristics | ANC0-7 | ANC8 | p-value[b] |
|---|---|---|---|
| 2nd trimester | 144 (69.6) | 63 (30.4) | |
| 3rd trimester | 18 (52.4) | 16 (47.6) | |

[a] Total of 2028 women were included in Table 2 after excluding total 413 women, which included 163 women who reported >16 antenatal contacts during the most recent pregnancy, and 250 woman missing the value on number of antenatal contacts.

[b] p-values based on design-based statistical tests: chi-square for categorical variables, adjusted Wald Test for continuous variables.

midwife only (42.3% vs 57.7%) or both doctor and nurse/midwife (22.1% vs 77.9%) compared to those with ANC8+. Importantly, greater proportion of women who initiated antenatal care contact in the second or third trimesters (6.6% and 52.4%, respectively) reported to be ANC0-7 than ANC8+.

Table 3 shows the crude prevalence ratios (cPrR) and adjusted prevalence ratios (aPrR) for ANC0-7 and respective 95% CIs. In crude model, ANC0-7 was associated with age, none or basic level of education, number of parities, wealth index, area of residence, type of ANC provider (either seeing a doctor only or nurse/midwife only) and timing of first contact for ANC ($P < 0.05$ for all). The prevalence of ANC0-7 increased as a number of parities increased and was higher among women in poorer or poorest wealth index quantile (aPrR 1.92; 95% CI 1.36, 2.72 for poorer and aPrR 1.75; 95% CI 1.10, 2.78 for poorest) compared to the richest women while controlling for age, education level, area of residence, type of ANC provider, and timing of first antenatal contact. The prevalence of ANC0-7, not meeting the WHO recommended number of antenatal contacts, was 48% and 72% higher among women living in North West Bank and South West Bank, respectively, compared to Central West Bank, while controlling for age, education level, parity, wealth index, and timing of initial antenatal contact. The women who delayed the first antenatal contact to the 2nd and 3rd trimesters had 2.80 times (95% CI 2.31, 3.39) and 2.72 times (95% CI 1.84, 4.03) the prevalence of those who completed the 1st antenatal contact in the 1st trimester. In adjusted model, age and education were not statistically significantly associated with ANC0-7.

## Discussion

Access to antenatal services enable women to receive appropriate and timely care to preserve health throughout gestation, which is expected to be coupled to adequate obstetric and postnatal care. Using the most recent 2019–2020 MICS data, our study analyzed responses of 2028 women on questions about ANC contacts during their most recent pregnancy within the previous two years. While ~95% of respondents reported receiving appropriate clinical assessments, including blood pressure measurement and collection of urine and blood tests, a quarter of women reported not meeting the WHO recommended number of ANC contacts, and 12% failed to initiate ANC contact in the first trimester. Those who did not meet the WHO recommendation of eight antenatal contacts (i.e., ANC0-7) were slightly older (28.7 vs 27.8 years), poorer, reported to be of higher parity, and less likely to see both doctor and nurse/midwife, and initiate an antenatal visit in the first trimester. Although the health systems in Palestine are challenged by recurrent conflicts, movement restrictions, insufficient resources and funding, shortage of health care providers, and complex politics, ANC coverage in 2019–2020 in Palestine was comparable to that in neighboring countries in the Middle Eastern Region [17, 21]. Based on the recent national household surveys in both Jordan and Egypt,

**Table 3. Crude and adjusted prevalence ratios (PrR) for ANC0-7 contacts among women who reported giving live birth within the last two years, 2019–2020 Palestinian Multiple Indicator Cluster Survey.**

| Women's characteristics | | Crude PrR[a] | Adjusted PrR[b] |
|---|---|---|---|
| Age | | 1.02 (1.01, 1.04)* | 0.98 (0.96, 1.00) |
| Women's education level | | | |
| Higher than Secondary | | 1.00 | 1.00 |
| Secondary | | 1.15 (0.97, 1.37) | 1.15 (0.80, 1.15) |
| None or Basic | | 1.23 (1.00, 1.51)* | 0.99 (0.78, 1.25) |
| Parity[c] | | 1.13 (1.09, 1.17)* | 1.39 (1.21, 1.59) |
| Wealth index | | | |
| | Richest | 1.00 | 1.00 |
| | Richer | 1.50 (1.11, 2.03)* | 1.45 (1.08, 1.95)* |
| | Middle | 1.57 (1.18, 2.09)* | 1.38 (1.04, 1.81)* |
| | Poorer | 1.76 (1.29, 2.41)* | 1.92 (1.36, 2.72)* |
| | Poorest | 1.59 (1.17, 2.17)* | 1.75 (1.10, 2.78)* |
| Area of residence | | | |
| | Central West Bank | 1.00 | 1.00 |
| | North West Bank | 1.50 (1.07, 2.09)* | 1.48 (1.07, 2.05)* |
| | South West Bank | 1.85 (1.34, 2.54)* | 1.72 (1.24, 2.38)* |
| | Gaza Strip | 1.61 (1.16, 2.24)* | 1.29 (0.84, 1.99) |
| Type of ANC provider, n (%) | | | |
| | Both Doctor and Nurse/Midwife | 1.00 | 1.00 |
| | Doctor only | 1.37 (1.08, 1.74)* | 1.78 (1.35, 2.35)* |
| | Nurse/Midwife only | 1.92 (1.36, 2.69)** | 1.72 (1.22, 2.44)* |
| Timing of 1st antenatal contact | | | |
| | 1st trimester | 1.00 | 1.00 |
| | 2nd trimester | 3.04 (2.58, 3.58)* | 2.80 (2.31, 3.39)* |
| | 3rd trimester | 2.23 (1.55, 3.37)* | 2.72 (1.84, 4.03)* |

*Prevalence ratios significant at P<0.05.

[a] Design-based univariable Poisson regression model was used to adjust for sample weight, clusters and stratum.

[b] Adjusted PrR was computed by design-based robust multivariable Poisson regression including sample weight, clusters, stratum, women's age, education level, quadratic term of parity, wealth index, area of residence, type of ANC provider, and timing of first antenatal contact.

[c] Quadratic term of parity was included in design-based robust multivariable Poisson regression to calculate the adjusted PrR of inadequate contact per unit increase in number of live births.

67.8% and 60.6% of women, respectively, initiated ANC contact in the first trimester and completed at least eight ANC visits [28].

Initiating care in the first trimester portended a higher probability of completing the WHO recommended number of ANC contacts. Mothers who delayed their visit beyond the first trimester were nearly three-fold less likely to comply to the WHO guideline than those presenting to clinics in the first trimester, after adjusting for age, education, parity, wealth index, and area of residence. The percentage of studied women not meeting the recommended number of visits increased with parity, such that women were 1.39 times less likely to achieve this WHO contact goal with each previously reported live birth, as has been seen elsewhere. Based on a secondary data analysis of 54 Demographic and Health Surveys and MICS conducted since 2012, Jiwani et al (2020) concluded that women with low wealth status were less likely to begin antenatal care on a timely basis [7]. The same study reported that women residing in larger

than smaller households, having shorter birth intervals and higher parity were also less likely to initiate early antenatal care and, thus, unlikely to achieve eight contacts [7]. Women with high parity may be less likely to seek ANC services because they may feel more confident with accumulated experience. Alternatively, it may be difficult for them to find time to seek antenatal services given their caretaker roles for children and other family members [29, 30]. In Palestine, there is considerable variation by location in the frequency with which pregnant women visit ANC clinics, with low contact likely a consequence of restricted movement imposed by blockades such as checkpoints, road gates, earthen walls, roadblocks, Separation Walls (barriers that cut through Palestinian towns in the West Bank) and bureaucratic or administrative restrictions on travel without permits [21, 31–33]. For example, women residing in the North and South regions of the West Bank are nearly twice as likely to not meet the WHO-recommended eight contacts than in the Central region. One study, using geographic spatial analysis, identified Palestinian communities in the West Bank as fragmented, like small "islands", that are separated and divided from each other while surrounded by settlement communities [34, 35]. Fragmentation of Palestinian lands can pose barriers to receiving healthcare, but also foster economic stagnation, limit access to education, degrade quality of life and increase exposure to political violence by the military and settlers [34]. In contrast, the Gaza Strip is a small area without checkpoints, with shorter distances to ANC facilities than in the West Bank [21]. While the population in the Gaza Strip faces difficult living conditions due to recurrent hostilities and ongoing siege, access to routine ANC contacts might have been easier in the Gaza Strip in 2019–2020 compared to the West Bank. However, the conflict that began on the 7th of October 2023 has severely destroyed health system infrastructure and impeded the movement of civilians, which have likely impeded access to and coverage of ANC in Palestine.

Political strife and conflict markedly disrupt provision and receipt of health care in many ways, such that supplies are reduced, retention of medical specialists is challenged, physical access to health care is interrupted by damage to infrastructure, and intensified fear for safety prevents people from seeking health care [21, 36]. The ongoing political instability, conflicts and violence, geographical separation, and deteriorating socioeconomic and living conditions continue to challenge Palestinian health systems [17, 21]. Practical approaches to continuously provide ANC services and strengthening its monitoring and evaluation mechanism in Palestine should be further considered.

A strength of this analysis is that it has drawn on representative data from the most recent 2019–2020 UNICEF Multiple Indicator Cluster Survey in Palestine, providing findings on antenatal care access that can be generalized to pregnant women living throughout Palestine. However, as a secondary analysis of a pre-existing survey, we had limited contextual influences to examine such as neighborhood characteristics and other local factors that could affect ANC coverage. Further, as with any survey collecting historical data, our results are prone to self-reporting recall bias, possibly leading to imprecise recall of number of ANC contacts and care received.

## Conclusions

The study results suggest that there was variation in achieving the WHO-recommended number of ANC contacts across socioeconomic status and clinic catchment areas. In particular, women with lower socioeconomic status may encounter obstacles in accessing ANC services, worsened by the geographic and political fragmentation in Palestine and mobility restrictions. There may be an opportunity to reduce inequalities to ANC coverage in Palestine; however, further studies on this issue and periodic evaluation of ANC services are likely needed to better inform and guide implementation of policies to improve ANC coverage in Palestine.

## Acknowledgments

The authors would like to acknowledge the George G. Graham Professorship Endowment in supporting the publication. The authors would like to thank Dr. Keith West, Jr. at Johns Hopkins Bloomberg School of Public Health by providing suggestions for the manuscript writing and organization. The first author greatly acknowledges the support from the United Nations Relief and Works Agency for Palestine Refugees in the Near East, the Ministry of Foreign Affairs of Japan, and from the Sight and Life Global Nutrition Research Institution, Baltimore, MD.

## Author Contributions

**Conceptualization:** Masako Horino, Saifuddin Ahmed, Yehia Abed.

**Formal analysis:** Masako Horino, Saifuddin Ahmed, Yehia Abed.

**Methodology:** Masako Horino, Saifuddin Ahmed, Khalid Abu Khalid, Yehia Abed.

**Supervision:** Saifuddin Ahmed, Yehia Abed.

**Writing – original draft:** Masako Horino, Saifuddin Ahmed.

**Writing – review & editing:** Masako Horino, Salwa Massad, Saifuddin Ahmed, Khalid Abu Khalid, Yehia Abed.

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
