## [Decision Letter · Decision Letter 0]

25 Jul 2023

PONE-D-23-16337Understanding coverage of antenatal care in Palestine: Cross-sectional Analysis of Palestinian Multiple Indicator Cluster Survey, 2019-2020PLOS ONE

Dear Dr. Horino,

Thank you for submitting your manuscript to PLOS ONE. After careful consideration, we feel that it has merit but does not fully meet PLOS ONE’s publication criteria as it currently stands. Therefore, we invite you to submit a revised version of the manuscript that addresses the points raised during the review process.

We look forward to receiving your revised manuscript.

Kind regards,

Veincent Christian Pepito

Academic Editor

PLOS ONE

Journal Requirements:

Additional Editor Comments:

Dear authors, thank you very much for your submission. Your submission looks promising, and I think you will be able to publish very soon. Kindly address the following comments in addition to the comments given by the reviewers:

1. I want you to incorporate a brief literature review on the topic to show what previous studies have been done on antenatal care in Palestine, or the determinants in antenatal care in general. This will strengthen the Introduction and Significance section of the paper as well as provides justification for the choice of variables that you will be including in your analysis. In connection to this, please provide justification for the variables that you have included in your analysis (i.e., why did you include these variables in your analysis? Which previous studies justify these choices?)

2. I appreciate the candid description of their methodology. However, I would like you to discuss your: (1) covariate selection strategy (i.e., how did you pick the variables to include in your regression model), and (2) any testing for overdispersion considering that you have used a Poisson model and the results of such testing. I am also of the opinion that one of the reason why your initial model failed to converge is that you are not supposed to use MLE methods for survey data.

3. Why not use Poisson regression for the univariate analysis as well? This is to ensure that crude and adjusted models are actually comparable.

4. Consider testing for departure from linearity assumption for quantitative variables like parity to reduce parameters estimated.

5. I found the Discussion section to be a little bit lacking especially on what you could recommend to address the issues that you have found. Please strengthen it by putting recommendations to address the issues that you have identified.

6. There are minor grammatical errors. Please have the manuscript reviewed by an English editor or a native speaker before resubmitting so that we can go straight to publication.

Reviewers' comments:

Reviewer's Responses to Questions

**Comments to the Author**

1. Is the manuscript technically sound, and do the data support the conclusions?

Reviewer #1: Yes

Reviewer #2: Yes

2. Has the statistical analysis been performed appropriately and rigorously? 

Reviewer #1: Yes

Reviewer #2: Yes

3. Have the authors made all data underlying the findings in their manuscript fully available?

Reviewer #1: Yes

Reviewer #2: Yes

4. Is the manuscript presented in an intelligible fashion and written in standard English?

Reviewer #1: Yes

Reviewer #2: Yes

5. Review Comments to the Author

Reviewer #1: Well done analyses, particularly commend use of prevalence ratios. Interesting and of course disturbing findings, although I note that overall ANC is higher than for example most of Africa. It might have been nice to compare ANC rates to other countries in the middle-eastern region as a normative comparison.

Reviewer #2: Is the manuscript technically sound, and do the data support the conclusions?

Yes. The presentation of results and discussion followed the objectives of the study. Conclusions were directly supported by the results of the study.Limitations such as recall bias were also explicitly stated in the Discussion and Conclusions.

Has the statistical analysis been performed appropriately and rigorously?

Yes. Details of the statistical analysis were elaborated and the results were presented in tables that are easy to follow and understand.

Have the authors made all data underlying the findings in their manuscript fully available?

Yes. The authors provided a link to download the MICS report/s and datasets (https://mics.unicef.org/surveys). However access to the datasets require registration as a MICS Data User from UNICEF.

Is the manuscript presented in an intelligible fashion and written in standard English? Yes. With some minor grammatical errors that need to be corrected prior to publication.

Line 39: Results: Overall, 28% of women did not *meet* the WHO recommendation of eight or more

Line 112: The *deficiency or lack of information* needed for management of ANC

Line 191: Table 1 summarizes the characteristics of 2,028 women aged between 15 to 49

*However, Table 1 in Line 208 indicates n= 2,191. Kindly reconcile the values.

6. PLOS authors have the option to publish the peer review history of their article (what does this mean?). If published, this will include your full peer review and any attached files.

Reviewer #1: **Yes: **Debra Jackson

Reviewer #2: No

---

## [Author Response · Author response to Decision Letter 0]

6 Sep 2023

Date: August 20, 2023

From: Masako Horino

Re: Submission ID: PONE-D-23-16337: “Understanding coverage of antenatal care in Palestine: Cross-sectional Analysis of Palestinian Multiple Indicator Cluster Survey, 2019-2020”

We are grateful to the editors and reviewers for their time and constructive comments on our manuscript. Please, find enclosed a revised manuscript addressing your comments and suggestions. We are enclosing two copies: one unmarked revised version and one where the most significant changes (mostly in response to your suggestions) have been highlighted. Below we provided point-by-point response to each of your and the reviewers’ comments. 

Editor’s Comments:

1. I want you to incorporate a brief literature review on the topic to show what previous studies have been done on antenatal care in Palestine, or the determinants in antenatal care in general. This will strengthen the Introduction and Significance section of the paper as well as provides justification for the choice of variables that you will be including in your analysis. In connection to this, please provide justification for the variables that you have included in your analysis (i.e., why did you include these variables in your analysis? Which previous studies justify these choices?)

Thank you for your comments. The following paragraph was included in the introduction section. 

“Previous studies have reported several factors associated with adequate antenatal care: initiation of care in the first trimester of pregnancy, residence in an urban area, especially close to an ANC facility, secondary or higher education, small household size (fewer than five members), higher socioeconomic status, lower parity (fewer than four live births), having health insurance, and being married vs single [3,7-10]. Furthermore, early pregnancy registration, clinical competence of the healthcare provider, and attentive interactions are also thought to be significant predictors of achieving the WHO recommended number of eight ANC contacts in pregnancy [7,10]. Globally, the percentage of women who initiate ANC in the first trimester has increased from 41% in 1999 to 59% in 2013. However, there is still a significant disparity in the receipt of any ANC coverage between low-income countries (24%) and high-income countries (82%) [5].”

2. I appreciate the candid description of their methodology. However, I would like you to discuss your: (1) covariate selection strategy (i.e., how did you pick the variables to include in your regression model), and (2) any testing for overdispersion considering that you have used a Poisson model and the results of such testing. I am also of the opinion that one of the reason why your initial model failed to converge is that you are not supposed to use MLE methods for survey data.

Thank you for your comments. For the first comment , we included the following sentence to the methodology section: “Then, a design-based multivariable log-binomial regression model was fitted to adjust for selected covariates that were identified to be the determinants of ANC coverage, based on the previous literature, including women’s age, education level, number of parities, wealth index, area of residence, type of ANC provider, and timing of first antenatal contact [3,7,9,10,25].”

For the second comment, we fitted a design-based robust Poisson model to relax the assumption of mean equals variance, and the estimated variances are larger than the conventional Poisson models under the simple random sampling assumption. 

3. Why not use Poisson regression for the univariate analysis as well? This is to ensure that crude and adjusted models are actually comparable.

We have checked analysis and the Poisson regression was used for the univariate analysis as suggested. The footnote on Table 3 was edited accordingly. 

4. Consider testing for departure from linearity assumption for quantitative variables like parity to reduce parameters estimated.

Thank you for your comment. As suggested, we tested for nonlinearity of age and parity by including both the continuous variables and quadratic terms in the design-based Poisson regression model. Since the quadratic term of age did not reach statistically significant level (p<0.05) suggesting that the relationship of age is linear, age was fitted as a continuous variable. However, both the quadratic term of parity and parity as a continuous variable were significantly associated with ANC coverage (p<0.05).

The following footnote was added to Table 3 to denote that a quadratic term of parity was included in the model: “c Quadratic term of parity was included in design-based robust multivariable Poisson regression to calculate the adjusted PrR of inadequate contact per unit increase in number of live births.”

5. I found the Discussion section to be a little bit lacking especially on what you could recommend to address the issues that you have found. Please strengthen it by putting recommendations to address the issues that you have identified.

Thank you for your comments. The following recommendation was added to the discission section. 

“To ensure that women in Palestine are adequately covered with ANC services, health facilities providing antenatal care in Palestine should update their guidelines per the updated WHO recommendation for the number of antenatal contacts and increase awareness of pregnant women on the importance of antenatal care in reducing the risk of adverse birth outcomes and pregnancy complications. Health facilities may consider using Short Message Service Alerts (SMSs) to ensure adherence to the WHO guidelines for the number of ANC visits [37]. Furthermore, continuous evaluation of ANC provision will benefit the Palestinian health systems by helping to prioritize the appropriate strategies, to meet the needs of Palestinian pregnant women, and to improve the quality of ANC services [16].”

6. There are minor grammatical errors. Please have the manuscript reviewed by an English editor or a native speaker before resubmitting so that we can go straight to publication.

Thank you for your comments. The authors have reviewed the English grammar carefully and made edits accordingly. 

Reviewers' comments:

Reviewer #1: Well done analyses, particularly commend use of prevalence ratios. Interesting and of course disturbing findings, although I note that overall ANC is higher than for example most of Africa. It might have been nice to compare ANC rates to other countries in the middle-eastern region as a normative comparison.

Thank you for your comment. We have included the following sentences in the discussion section. 

“Although the health systems in Palestine are challenged by recurrent conflicts, movement restrictions, insufficient resources and funding, shortage of health care providers, and complex politics, ANC coverage in Palestine is comparable to that in neighboring countries in the Middle Eastern Region [17,21]. Based on the recent national household surveys in both Jordan and Egypt, 67.8% and 60.6% of women, respectively, initiated ANC contact in the first trimester and completed at least eight ANC visits [28].” 

Reviewer #2: Is the manuscript technically sound, and do the data support the conclusions?

Yes. The presentation of results and discussion followed the objectives of the study. Conclusions were directly supported by the results of the study. Limitations such as recall bias were also explicitly stated in the Discussion and Conclusions.

Is the manuscript presented in an intelligible fashion and written in standard English? 

Yes. With some minor grammatical errors that need to be corrected prior to publication.

Line 39: Results: Overall, 28% of women did not *meet* the WHO recommendation of eight or more

Corrected.

Line 112: The *deficiency or lack of information* needed for management of ANC

Corrected.

Line 191: Table 1 summarizes the characteristics of 2,028 women aged between 15 to 49

*However, Table 1 in Line 208 indicates n= 2,191. Kindly reconcile the values.

 Corrected.

We hope that the above responses and the enclosed revised version of the manuscript properly address the concerns of the reviewers; but please, let us know if you have any additional questions or concerns.

Sincerely,

Masako Horino, representing all coauthors

---

## [Decision Letter · Decision Letter 1]

24 Oct 2023

PONE-D-23-16337R1Understanding coverage of antenatal care in Palestine: Cross-sectional Analysis of Palestinian Multiple Indicator Cluster Survey, 2019-2020PLOS ONE

Dear Dr. Horino,

Thank you for submitting your manuscript to PLOS ONE. After careful consideration, we feel that it has merit but does not fully meet PLOS ONE’s publication criteria as it currently stands. Therefore, we invite you to submit a revised version of the manuscript that addresses the points raised during the review process.

We look forward to receiving your revised manuscript.

Kind regards,

Veincent Christian Pepito

Academic Editor

PLOS ONE

Journal Requirements:

Additional Editor Comments:

Thanks for working on this. May I clarify why variables in Table 1 do not add up to 2028?

Also please add brief commentary on the current issue in Palestine. Consider the comments of Reviewer 2 as well.

Reviewers' comments:

Reviewer's Responses to Questions

**Comments to the Author**

1. If the authors have adequately addressed your comments raised in a previous round of review and you feel that this manuscript is now acceptable for publication, you may indicate that here to bypass the “Comments to the Author” section, enter your conflict of interest statement in the “Confidential to Editor” section, and submit your "Accept" recommendation.

Reviewer #1: All comments have been addressed

Reviewer #2: All comments have been addressed

2. Is the manuscript technically sound, and do the data support the conclusions?

Reviewer #1: Yes

Reviewer #2: Partly

3. Has the statistical analysis been performed appropriately and rigorously? 

Reviewer #1: Yes

Reviewer #2: Yes

4. Have the authors made all data underlying the findings in their manuscript fully available?

Reviewer #1: Yes

Reviewer #2: Yes

5. Is the manuscript presented in an intelligible fashion and written in standard English?

Reviewer #1: Yes

Reviewer #2: Yes

6. Review Comments to the Author

Reviewer #1: Well done revision. I was impressed with the responses to the complex statistical questions and the adding of footnotes to explain is a welcome addition.

Reviewer #2: Kindly review the recommendations such that each point is supported by the results/findings in the study

1) health facilities should update their guidelines per the updated WHO recommendations on the number of ANC contacts: There is no explicit mention in the study about the existing local policies on antenatal care services which should be the basis of the recommendation to update the local guidelines.

2) health facilities may consider using SMS to ensure adherence to WHO guidelines: There is no explicit mention in the study about women's access to telecommunication services, nor the acceptability and availability of these services for the women and their families. This recommendation should also consider the capacity of the health facilities to provide SMS services. Recommendations should be supported by the findings of the study.

3) continuous evaluation of ANC provision: The study mentioned in its introduction that there is "little reported national data" and "lack of information needed" which can be presumed to be related to a lack of accessible administrative reports on the provision of health services including ANC at the national scale. As such, it seems that there is a need to establish or strengthen the monitoring and evaluation mechanism for the provision of ANC services instead of "continuously evaluating" its provision.

7. PLOS authors have the option to publish the peer review history of their article (what does this mean?). If published, this will include your full peer review and any attached files.

Reviewer #1: **Yes: **Debra Jackson

Reviewer #2: No

---

## [Author Response · Author response to Decision Letter 1]

9 Dec 2023

Date: December 9, 2023

From: Masako Horino

Re: Submission ID: PONE-D-23-16337: “Understanding coverage of antenatal care in Palestine: Cross-sectional Analysis of Palestinian Multiple Indicator Cluster Survey, 2019-2020”

We would like to thank the editors and reviewers for their time and constructive comments on our manuscript. Please, find enclosed a revised manuscript addressing your comments and suggestions. We are enclosing two copies: one unmarked revised version and one where the most significant changes (mostly in response to your suggestions) have been highlighted. Below we provided point-by-point response to each of your and the reviewers’ comments. 

Additional Editor’s Comments:

Thanks for working on this. May I clarify why variables in Table 1 do not add up to 2028?

Thank you for your comment. The variables in Table 1 did not add up to 2028, because the analysis was weighted by the original (non-normalized) weight. However, we have now applied normalized weights to match the analytical total sample size to the observed total sample size, which we hope will avoid confusion in the sample size. 

Also please add brief commentary on the current issue in Palestine. Consider the comments of Reviewer 2 as well.

Thank you for your comment. The following sentence has been included in the discussion section (lines 332-334). 

“However, the conflict that began on the 7th of October 2023 has severely destroyed health system infrastructure in the Gaza Strip and limited the movement of civilians in the West Bank, which have likely impeded access to and coverage of ANC in Palestine.”

Reviewer #2s' comments:

1) health facilities should update their guidelines per the updated WHO recommendations on the number of ANC contacts: There is no explicit mention in the study about the existing local policies on antenatal care services which should be the basis of the recommendation to update the local guidelines.

Thank you for your comment. We have decided not to include this recommendation. 

2) health facilities may consider using SMS to ensure adherence to WHO guidelines: There is no explicit mention in the study about women's access to telecommunication services, nor the acceptability and availability of these services for the women and their families. This recommendation should also consider the capacity of the health facilities to provide SMS services. Recommendations should be supported by the findings of the study.

Thank you for your comment. We have decided not to include this recommendation. 

3) continuous evaluation of ANC provision: The study mentioned in its introduction that there is "little reported national data" and "lack of information needed" which can be presumed to be related to a lack of accessible administrative reports on the provision of health services including ANC at the national scale. As such, it seems that there is a need to establish or strengthen the monitoring and evaluation mechanism for the provision of ANC services instead of "continuously evaluating" its provision.

Thank you for your comment. The following sentence has been added to the discussion section (lines 340-343).

“Practical approaches to continuously provide ANC services and strengthening its monitoring and evaluation mechanism in Palestine should be further considered.”

We hope that the above responses and the enclosed revised version of the manuscript properly address the concerns of the reviewers; but please, let us know if you have any additional questions or concerns.

Sincerely,

Masako Horino, representing all coauthors

---

## [Editor Report · Decision Letter 2]

16 Jan 2024

Understanding coverage of antenatal care in Palestine: Cross-sectional Analysis of Palestinian Multiple Indicator Cluster Survey, 2019-2020

PONE-D-23-16337R2

Dear Dr. Horino,

We’re pleased to inform you that your manuscript has been judged scientifically suitable for publication and will be formally accepted for publication once it meets all outstanding technical requirements.

Kind regards,

Veincent Christian Pepito

Academic Editor

PLOS ONE

Additional Editor Comments (optional):

Thanks for your re-submission and congratulations on your new paper!
---

## [Editor Report · Acceptance letter]

25 Jan 2024

PONE-D-23-16337R2 

PLOS ONE

Dear Dr. Horino, 

I'm pleased to inform you that your manuscript has been deemed suitable for publication in PLOS ONE. Congratulations! Your manuscript is now being handed over to our production team.

Kind regards, 

on behalf of

Mr Veincent Christian Pepito 

Academic Editor

PLOS ONE